# MoRA: High-Rank Updating for Parameter-Efficient Fine-Tuning

## Abstract

Low-rank adaptation (LoRA) is a popular parameter-efficient fine-tuning (PEFT) method for large language models (LLMs). In this paper, we analyze the impact of low-rank updating, as implemented in LoRA. Our findings suggest that the low-rank updating mechanism may limit the ability of LLMs to effectively learn and memorize new knowledge. Inspired by this observation, we propose a new method called MoRA, which employs a square matrix to achieve high-rank updating while maintaining the same number of trainable parameters. To achieve it, we introduce the corresponding non-parameter operators to reduce the input dimension and increase the output dimension for the square matrix. Furthermore, these operators ensure that the weight can be merged back into LLMs, which enables our method to be deployed like LoRA. We perform a comprehensive evaluation of our method across five tasks: instruction tuning, mathematical reasoning, continual pretraining, memory and pretraining. Our method outperforms LoRA on memory-intensive tasks and achieves comparable performance on other tasks.

## 1 Introduction

As the size of language models increases, parameter-efficient fine-tuning (PEFT) Houlsby et al. (2019) has emerged as a popular technique to adapt these models to specific downstream tasks. Compared to Full Fine-Tuning (FFT), which updates all model parameters, PEFT modifies only a small part of the parameters. For example, it can achieve similar performance with FFT by updating less than 1% of the parameters in some tasks Hu et al. (2021), which significantly reduces the memory requirements for the optimizer and facilitates the storage and deployment of fine-tuned models.

Among the existing PEFT methods, Low-Rank Adaptation (LoRA) Hu et al. (2021) is particularly prevalent for LLMs. LoRA enhances performance over other PEFT methods such as prompt tuning Lester et al. (2021) or adapters Houlsby et al. (2019) by updating parameters via low-rank matrices. These matrices can be merged into the original model parameters, thereby avoiding additional computational costs during inference. There are numerous methods that aim to improve LoRA for LLMs. However, most methods primarily validate their efficiency based on GLUE Wang et al. (2018), either by achieving better performance or by requiring fewer trainable parameters. Recent methods Liu et al. (2024); Meng et al. (2024); Zhu et al. (2024) leverage instruction tuning task such as Alpaca Wang et al. (2024) or reasoning tasks like GSM8K Cobbe et al. (2021) to better evaluate their performance on LLMs. However, the diverse settings and datasets used in the evaluation complicate the understanding of their progression.

In this paper, we conduct a comprehensive evaluation of LoRA across various tasks under the same settings, including instruction tuning, mathematical reasoning, and continual pretraining. We find that LoRA-like methods demonstrate similar performance across these tasks and they perform comparably to FFT in instruction tuning but fall short in mathematical reasoning and continual pretraining. Among these tasks, instruction tuning primarily focuses on interacting with the format, rather than acquiring knowledge and capabilities, which are learned almost entirely during pretraining Zhou et al. (2024). We observe that LoRA is easily adapted to follow response formats in instruction tuning but struggles with other tasks that require enhancing knowledge and capabilities through fine-tuning.

One plausible explanation for this limitation observed with LoRA could be its reliance on low-rank updates Lialin et al. (2023). The low-rank update matrix, $\Delta W$, struggles to estimate the full-rank updates in FFT, particularly in memory-intensive tasks like continual pretraining that require memorizing domain-specific knowledge. Since the rank of $\Delta W$ is significantly smaller than the full rank, this limitation restricts capacity to store new information via fine-tuning. Moreover, current variants of LoRA cannot alter the inherent characteristic of low-rank updates. To validate this, we conducted a memorization task using pseudo-data to assess the performance of LoRA in memorizing new knowledge. We found that LoRA performed significantly worse than FFT, even with a large rank such as 256.

Given these observations, we introduce a method called MoRA, which employs a square matrix as opposed to low-rank matrices, aiming to maximize the rank in $\Delta W$ while maintaining the same number of trainable parameters. For instance, when utilizing 8 rank with the hidden size 4096, LoRA employs two low-rank matrices $A \in \mathbb{R}^{4096 \times 8}$ and $B \in \mathbb{R}^{8 \times 4096}$, with $rank(\Delta W) \leq 8$. Under same number of parameters, our method uses a square matrix $M \in \mathbb{R}^{256 \times 256}$, with $rank(\Delta W) \leq 256$, as depicted in Figure 1. Notably, our method exhibits a greater capacity than LoRA with a large rank. To decrease the input dimension and increase the output dimension for $M$, we develop corresponding non-parameter operators. Furthermore, these operators and $M$ can be substituted by a $\Delta W$, ensuring our method can be merged back into LLM like LoRA.

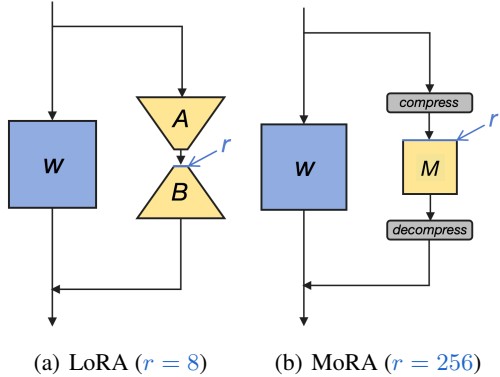

(a) LoRA ($r = 8$)   (b) MoRA ($r = 256$)

Figure 1: An overview of our method compared to LoRA under **same** number of trainable parameters. $W$ is the frozen weight from model. r represents the rank in two methods.

Our contributions are as follows:

1. We introduce MoRA, a novel method that employs a square matrix instead of low-rank matrices in LoRA to achieve high-rank updating, while maintaining the same number of trainable parameters.

2. We discuss four kinds of non-parameter operators of MoRA to reduce the input dimension and increase the output dimension for the square matrix, while ensures that the weight can be merged back into LLMs.

3. We evaluate MoRA across five tasks: memory, instruction tuning, mathematical reasoning, continual pretraining, and pretraining. Our method outperforms LoRA on memory-intensive tasks and achieves comparable performance on other tasks, which demonstrates the effectiveness of high-rank updating.

## 2 RELATED WORK

### 2.1 LORA

LoRA is one of the most popular PEFT methods for fine-tuning LLM, owing to its broad applicability and robust performance in comparison to other methods. To approximate the updated weight $\Delta W$ in FFT, LoRA employs two low-rank matrices for its decomposition. By adjusting the rank of these two matrices, LoRA can accordingly modify the trainable parameters. As a result, LoRA can merge these matrices after fine-tuning without incurring the inference latency associated with FFT. There are many methods to further improve LoRA, particularly for the application in LLMs. DoRALiu et al. (2024) further decomposes the original weight into magnitude and direction components and uses LoRA to update the direction component. LoRA+Hayou et al. (2024) employs different learning rates for the two low-rank matrices to improve learning efficiency. ReLoRALialin et al. (2023) integrates LoRA into the LLM during training to increase the rank of the final $\Delta W$.

## 2.2 Fine-Tuning with LLMs

Despite the impressive performance of LLMs with in-context learning, certain scenarios still necessitate fine-tuning, which can be broadly categorized into three types. The first type, instruction tuning, aims to better align LLMs with end tasks and user preferences, without significantly enhancing the knowledge and capabilities of LLMs Zhou et al. (2024). This approach simplifies the process of dealing with varied tasks and understanding complex instructions. The second type involves complex reasoning tasks such as mathematical problem-solving Collins et al. (2023); Imani et al. (2023); Yu et al. (2023), where general instruction tuning often falls short in handling complex, symbolic, multi-step reasoning tasks. To improve the reasoning abilities of LLMs, the majority of research focuses on creating corresponding training datasets, either by leveraging larger teacher models like GPT-4 Fu et al. (2023), or by rephrasing questions along a reasoning path Yu et al. (2023). The third type, continual pretraining Cheng et al. (2023); Chen et al. (2023); Han et al. (2023); Liu et al. (2023), aims to enhance the domain-specific capabilities of LLMs. Unlike instruction tuning, it necessitates fine-tuning to augment the corresponding domain-specific knowledge and capabilities.

However, most variants of LoRA Kopiczko et al. (2023); Lialin et al. (2023); Dettmers et al. (2024); Zhu et al. (2024) predominantly employ instruction tuning or text classification tasks from GLUE Wang et al. (2018) to validate their efficacy on LLMs. Given that instruction tuning requires the least capacity for fine-tuning compared to other types, it may not accurately reflect the effectiveness of LoRA variants. To better evaluate their methods, recent works Meng et al. (2024); Liu et al. (2024); Shi et al. (2024); Renduchintala et al. (2023) have employed reasoning tasks to test their methods. But the training sets used are often too small for LLMs to effectively learn reasoning. For instance, some methods Meng et al. (2024); Renduchintala et al. (2023) utilize the GSM8K Cobbe et al. (2021) with only 7.5K training samples. Compare to the SOTA method with 395K training samples Yu et al. (2023), this small training set achieves worse performance on reasoning and makes it hard to evaluate the effectiveness of these methods.

## 3 Analysis the Influence of Low-rank Updating

The key idea of LoRA Hu et al. (2021) involves the use of low-rank updates to estimate full-rank updates in FFT. Formally, given a pretrained parameter matrix $W_0 \in \mathbb{R}^{d \times k}$, LoRA employs two low-rank matrices to calculate the weight update $\Delta W$:

$$h = W_0 x + \Delta W x = W_0 x + BA x \qquad (1)$$

where $A \in \mathbb{R}^{r \times k}$ and $B \in \mathbb{R}^{d \times r}$ represent the low-rank matrices in LoRA. To ensure that $\Delta W = 0$ at the beginning of training, LoRA initializes $A$ with a Gaussian distribution and $B$ with zero. Due to the low-rank decomposition of $\Delta W$ into $BA$, the $rank(\Delta W) \leq r$. The weight update in LoRA exhibits a markedly low rank, $r \ll \min(d, k)$, in comparison to the full-rank updating in FFT. Low-rank updating by LoRA shows on-par performance with full-rank updating in some tasks such as text classification or instruction tuning Liu et al. (2024); Meng et al. (2024). However, for tasks like complex reasoning or continual pretraining, LoRA tends to show worse performance Liu et al. (2023).

Based on these observations, we hypothesize that low-rank updating easily leverages the original knowledge and capabilities of LLMs to solve tasks but struggles with tasks that require enhancing the knowledge and capabilities of LLMs.

To substantiate this hypothesis, we examine the differences between LoRA and FFT in terms of memorizing new knowledge through fine-tuning. In order to circumvent leveraging the original knowledge of the LLM, we randomly generate 10K pairs of Universally Unique Identifiers (UUIDs), each pair comprising two UUIDs with 32 hexadecimal values. The task requires the LLM to generate the corresponding UUID based on the input UUID. For instance, given a UUID such as "205f3777-52b6-4270-9f67-c5125867d358", the model should generate the corresponding UUID based on 10K training pairs. This task can also be viewed as a question-answering task, while the knowledge indispensable for accomplishing it is exclusively from the training datasets rather than the LLM itself.

For the training settings, we employ LLaMA-2 7B as base model, utilizing 1,000 pairs per batch and conducting 100 epochs. For the LoRA, we apply low-rank matrices to all linear layers and search learning rate from {1e-4,2e-4,3e-4} to enhance performances.

We conduct the experiment on LoRA using various ranks $r \in \{8, 16, 32, 64, 128, 256\}$. For the FFT, we directly use a learning rate of 3e-5. Based on Figure 2, we observe low-rank updating are hard to memorizing new knowledge compared to FFT. Although constantly increasing the rank of LoRA can alleviate this problem, the gap still exists.

In contrast to the memory task, we also evaluate the performance gap between LoRA and FFT on instruction tuning, which merely introduces new knowledge. Similar to previous results Meng et al. (2024); Zhu et al. (2024), we also find that LoRA matches the performance of FFT with small rank $r = 8$ in Table 2. This indicates that LoRA can easily leverage the original knowledge of LLMs by fine-tuning like FFT.

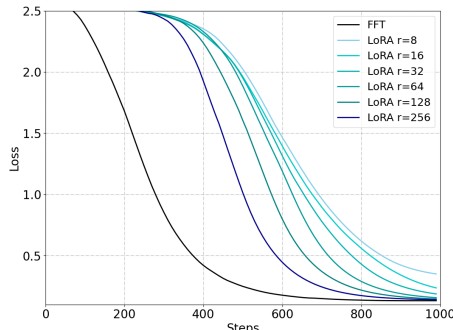

Figure 2: Performance of memorizing UUID pairs through fine-tuning with FFT and LoRA.

## 4 METHOD

Based on the above analysis, we propose a new method to alleviate the negative effects of low-rank updating. The main idea of our method is to utilize the same trainable parameters as much as possible to achieve a higher rank in $\Delta W$. Consider the pretrained weight $W_0 \in \mathbb{R}^{d \times k}$. LoRA uses two low-rank matrices $A$ and $B$ with $(d + k)r$ total trainable parameters for rank $r$. Under same trainable parameters, a square matrix $M \in \mathbb{R}^{\hat{r} \times \hat{r}}$ where $\hat{r} = \lfloor \sqrt{(d + k)r} \rfloor$ can achieve the highest rank due to $r \ll \min(d, k)$.

To accomplish this, we need to reduce the input dimension and increase the output dimension for $M$. Formally,

$$h = W_0 x + f_{\text{decomp}} \left( M f_{\text{comp}} \left( x \right) \right) \tag{2}$$

where $f_{\text{comp}} : \mathbb{R}^k \to \mathbb{R}^{\hat{r}}$ denotes the function that decreases the input dimension of $x$ from $k$ to $\hat{r}$, and $f_{\text{decomp}} : \mathbb{R}^{\hat{r}} \to \mathbb{R}^d$ represents the function that enhances the output dimension from $\hat{r}$ to $d$. Furthermore, these two functions ought to be non-parameterized operators and expected to execute in linear time corresponding to the dimension. They should also have corresponding function, $f_{\overline{\text{comp}}} : \mathbb{R}^{\hat{r} \times \hat{r}} \to \mathbb{R}^{\hat{r} \times k}$ and $f_{\overline{\text{decomp}}} : \mathbb{R}^{\hat{r} \times k} \to \mathbb{R}^{d \times k}$, to transform $M$ into $\Delta W$. For any $x$, the following should hold:

$$f_{\text{decomp}} \left( M f_{\text{comp}} \left( x \right) \right) = \Delta W x, \forall x \in \mathbb{R}^k \tag{3}$$

where $\Delta W = f_{\overline{\text{decomp}}} (f_{\overline{\text{comp}}} (M))$. If Eq. 3 holds, $M$ can be losslessly expanded to $\Delta W$ based on $f_{\text{comp}}$ and $f_{\text{decomp}}$. This allows our method to merge back into the LLM like LoRA.

For the design of $f_{\text{comp}}$ and $f_{\text{comp}}$, we explore several methods to implement these functions. One straightforward method is truncating the dimension and subsequently add it in corresponding dimension. Formally, this can be represented as:

$$f_{\text{comp}} \left( x \right) = x_{1:\hat{r}}$$
$$f_{\text{decomp}} \left( x \right) = \begin{bmatrix} x \\ \mathbf{0} \end{bmatrix} \tag{4}$$

and the corresponding $\Delta W$ is:

$$\Delta W = \begin{bmatrix} M & \mathbf{0} \\ \mathbf{0} & \mathbf{0} \end{bmatrix} \tag{5}$$

However, this method leads to a significant loss of information during compression and only modifies a segment of the output by appending a zero vector during decompression. To improve it, we can share the rows and columns of $M$ to achieve a more efficient compression and decompression. Formally, this can be represented as:

$$f_{\text{comp}} \left( x \right) = \left[ \sum_{j \in g_i} x_j \right]_{i=1}^r$$
$$f_{\text{decomp}} \left( x \right) = \left[ x_{\widetilde{g}'_i} \right]_{i=1}^d \tag{6}$$

Here, $g$ and $g'$ represent predefined groups that share the same row and column in $M$, respectively. The $j \in g_i$ indicates that the $j$-th dimension belongs to the $i$-th group in $g$. The term $\widetilde{g}'_i$ is the reverse of $g'_i$, referring to the $i$-th dimension associated with the $\widetilde{g}'_i$-th group in $g'$. The corresponding $\Delta W$ is as follows:

$$\Delta W_{i,j} = M_{\widetilde{g}'_i, \widetilde{g}_j} \tag{7}$$

Sharing rows and columns can be efficient for larger ranks such as $r = 128$ or $r = 256$, as only a few rows or columns in $\Delta W$ share a common row or column. For instance, considering to $\Delta W \in \mathbb{R}^{4096 \times 4096}$ for $r = 128$, which has $\hat{r} = 1024$ and $M \in \mathbb{R}^{1024 \times 1024}$. In this situation, only 4 rows or columns share the same row or column. Conversely, for smaller ranks such as $r = 8$, where $\hat{r} = 256$, it requires average 16 rows or columns in a group to share the same row or column in $M$. It can lead to inefficiencies due to the significant information loss during compression in Eq. 6.

To enhance performance for smaller ranks, we reshape $x$ instead of directly compressing it, to preserve the input information. In this context, $f_{\text{comp}}(x) : \mathbb{R}^k \to \mathbb{R}^{n \times \hat{r}}$ and $f_{\text{decomp}} : \mathbb{R}^{n \times \hat{r}} \to \mathbb{R}^d$. Corresponding $f_{\text{comp}}$, $f_{\text{decomp}}$ and $\Delta W$ are as follows:

$$f_{\text{comp}}(x) = \begin{bmatrix} x_{1:\hat{r}} & x_{\hat{r}:2\hat{r}} & \cdots & x_{(n-1)\hat{r}:n\hat{r}} \end{bmatrix}$$

$$f_{\text{decomp}}(x) = \text{concat}(x)$$

$$\Delta W = \begin{bmatrix} M & \mathbf{0} & \cdots & \mathbf{0} \\ \mathbf{0} & M & \cdots & \mathbf{0} \\ \vdots & \vdots & \ddots & \vdots \\ \mathbf{0} & \mathbf{0} & \cdots & M \end{bmatrix} \tag{8}$$

where $\text{concat}(x)$ refers to concatenate the rows of $x$ into a vector. For simplicity, we omit the padding and truncation operators in above functions and focus on the case where $d = k$. In comparison to sharing columns and rows, this method incurs additional computational overhead by reshaping $x$ into $\mathbb{R}^{n \times \hat{r}}$ instead of $\mathbb{R}^{\hat{r}}$. However, given that the size of $M$ is significantly smaller than $W_0$, this additional computation is very small for rank like 8. For instance, when fine-tuning the 7B model with rank of 8 ($\hat{r} = 256$), this method is only 1.03 times slower than the previous methods.

Inspired by RoPE Su et al. (2024), we can further refine this method by incorporating rotation operators into $f_{\text{comp}}$ to augment the expressiveness of $M$ by enable it to differentiate between various $x_{i\hat{r}:(i+1)\hat{r}}$ by rotating them. We can modify Eq. 8 as follows:

$$f_{\text{comp}}(x) = \begin{bmatrix} a^1 & a^2 & \cdots & a^{n-1} \end{bmatrix}$$

$$\Delta W = \begin{bmatrix} P^1 & \mathbf{0} & \cdots & \mathbf{0} \\ \mathbf{0} & P^2 & \cdots & \mathbf{0} \\ \vdots & \vdots & \ddots & \vdots \\ \mathbf{0} & \mathbf{0} & \cdots & P^{n-1} \end{bmatrix} \tag{9}$$

where $a^i$ and $P^i$ represent the corresponding values of $x_{i\hat{r}:(i+1)\hat{r}}$ and $M$ post-rotation, respectively. Following RoPE, we use a $\hat{r} \times \hat{r}$ block diagonal matrix to achieve the rotation. However, our method use rotation information to enable $M$ distinguish the $x_{i\hat{r}:(i+1)\hat{r}}$ instead of token position in RoPE. We can define $a^i$ and $P^i$ as follows:

$$a^i = \begin{bmatrix} R_{\theta_1,i} & \mathbf{0} & \cdots & \mathbf{0} \\ \mathbf{0} & R_{\theta_2,i} & \cdots & \mathbf{0} \\ \vdots & \vdots & \ddots & \vdots \\ \mathbf{0} & \mathbf{0} & \cdots & R_{\theta_{\frac{\hat{r}}{2}},i} \end{bmatrix} x_{i\hat{r}:(i+1)\hat{r}}$$

$$P^i = M \begin{bmatrix} R_{\theta_1,i} & \mathbf{0} & \cdots & \mathbf{0} \\ \mathbf{0} & R_{\theta_2,i} & \cdots & \mathbf{0} \\ \vdots & \vdots & \ddots & \vdots \\ \mathbf{0} & \mathbf{0} & \cdots & R_{\theta_{\frac{\hat{r}}{2}},i} \end{bmatrix} \tag{10}$$

where $\theta_j = 10000^{-2(j-1)/\hat{r}}$ and $R_{\theta_j,i} \in \mathbb{R}^{2 \times 2}$ is a rotation matrix:

$$R_{\theta_j,i} = \begin{bmatrix} \cos i\theta_j & -\sin i\theta_j \\ \sin i\theta_j & \cos i\theta_j \end{bmatrix} \tag{11}$$

## 5 EXPERIMENT

We evaluate our method on various tasks to understand the influence of high-rank updating. In Section 5.1, we evaluate our method with LoRA and our method on memorizing UUID pairs to show the benefit of high-rank updating on memorizing. In Section 5.2, we reproduce LoRA, LoRA variants and FFT on three fine-tuning tasks: instruction tuning, mathematical reasoning and continual pretraining. In Section 5.3, we compare our method with LoRA and ReLoRA on pretraining by training transformer from scratch.

### 5.1 MEMORIZING UUID PAIRS

We first compare our method with LoRA and FFT on memorizing UUID pairs to demonstrate improvements through high-rank updating. Following the training settings in Section 3, we search learning rate from {5e-5,7e-5,1e-4} and use decompress and compress functions in Eq. 8, sharing rows and columns in $M$. Due to use one matrix $M$ instead of two matrices $A$ and $B$, we can directly initialize $M$ with zeros. For the predefined groups $g$ and $g'$, we group every adjacent $\hat{r}$ rows or columns together. The training loss is presented in Figure3. Our method shows significant improvements over LoRA with the same number of trainable parameters, benefiting from high-rank updating. We also report character-level accuracy at various training steps in Table 1. MoRA requires fewer training steps to memorize these UUID pairs compared to LoRA. Compared to FFT, MoRA with 256 rank can achieve similar performance and both method can memorize all UUID pairs in 500 steps.

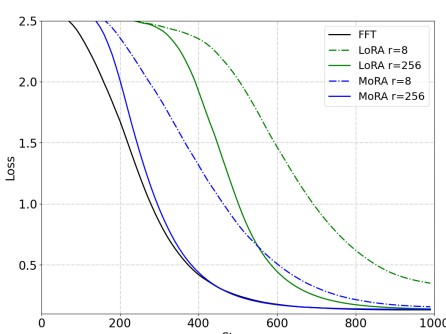

|  | Rank | 300 | 500 | 700 | 900 |
|---|---|---|---|---|---|
| FFT | - | 42.5 | 100 | 100 | 100 |
| LoRA | 8 | 9.9 | 10.0 | 10.7 | 54.2 |
| MoRA | 8 | 10.1 | 15.7 | 87.4 | 100 |
| LoRA | 256 | 9.9 | 70.6 | 100 | 100 |
| MoRA | 256 | 41.6 | 100 | 100 | 100 |

Table 1: Character-level accuracy of memorizing UUID pairs by generating the value of corresponding key in 300, 500, 700 and 900 training steps.

Figure 3: Performance of memorizing UUID pairs with LoRA and our method on rank 8 and 256.

### 5.2 FINE-TUNING TASKS

#### 5.2.1 SETUP

We evaluate our method across three fine-tuning tasks for large language models (LLMs): instruction tuning, mathematical reasoning, and continual pretraining. For these tasks, we select high-quality corresponding datasets to test both LoRA and our method. In instruction tuning, we utilize Tülu v2 Ivison et al. (2023), a blend of several high-quality instruction datasets, containing 326k filtered samples. We assess instruction performance using the MMLU Hendrycks et al. (2020) in both zero-shot and five-shot settings. For mathematical reasoning, we employ the MetaMath Yu et al. (2023) with its 395k samples to enhance mathematical reasoning capabilities and also use GSM8K Cobbe et al. (2021) and MATH Hendrycks et al. (2021) for further evaluation. In continual pretraining, we adapt an LLM to the biomedicine and finance using PubMed abstracts from the Pile Gao et al. (2020) and finicial news, complemented by data preprocessing methods from AdaptLLM Cheng et al. (2023) to boost performance. We report the average performance of corresponding tasks for continual pretraining. More details can be found in Appendix C.

| | | Instruction Tuning | | Mathematical Reasoning | | Continual Pretraining | |
|---|---|---|---|---|---|---|---|
| **Method** | **Rank** | **MMLU 0** | **MMLU 5** | **GSM8K** | **MATH** | **BioMed.** | **Finance** |
| FFT | - | 50.6 | 51.3 | 66.6 | 20.1 | 56.4 | 69.6 |
| LoRA | 8 | 50.2 | 51.5 | **64.6** | 15.1 | 52.3 | 64.0 |
| LoRA+ | 8 | 49.2 | 51.1 | 64.1 | **15.8** | 52.2 | 64.9 |
| ReLoRA | 8 | 49.3 | 50.2 | 61.5 | 14.5 | 46.3 | 61.0 |
| AsyLoRA | 8 | **50.3** | **52.2** | 64.5 | 15.0 | 52.5 | 63.5 |
| DoRA | 8 | 50.2 | 51.5 | 64.5 | 14.6 | 52.5 | 63.9 |
| MoRA (Ours) | 8 | 49.7 | 51.5 | 64.2 | 15.4 | **53.3** | **67.1** |
| LoRA | 256 | 49.7 | 50.8 | 67.9 | **19.9** | 54.1 | 67.3 |
| LoRA+ | 256 | 49.2 | 51.3 | **68.2** | 17.1 | 54.2 | 66.7 |
| ReLoRA | 256 | - | - | 64.0 | 18.1 | 52.9 | 57.9 |
| AsyLoRA | 256 | **50.1** | **52.0** | 66.9 | 19.3 | 54.1 | 66.9 |
| DoRA | 256 | 49.6 | 51.1 | 67.4 | 19.5 | 54.2 | 66.0 |
| MoRA (Ours) | 256 | 49.9 | 51.4 | 67.9 | 19.2 | **55.4** | **68.7** |

Table 2: Performance of FFT, LoRA, LoRA variants and our method on instruction tuning, mathematical reasoning and continual pretraining tasks.

### 5.2.2 BASELINES AND IMPLEMENTS

For LoRA-like methods and MoRA, we conducted experiments at $r = 8$ and $r = 256$, and reproduce following methods across three tasks: FFT, LoRA, LoRA+ Hayou et al. (2024), AsyLoRA Zhu et al. (2024), ReLoRA Lialin et al. (2023) and DoRA Liu et al. (2024). LoRA+ enhances the learning rate of matrix $B$ in LoRA to facilitate efficient feature learning based on theoretical analysis. We search the corresponding the hyperparameter $\lambda$ from $\{2,4\}$. AsyLoRA also analyzes asymmetry in the $A$ and $B$ matrices, and we adopted their initialization strategy. ReLoRA proposes a method to merge low-rank matrices into the model during training to increase the rank of $\Delta W$. we search merge steps from $\{1k, 2k\}$ and use 50 steps restarts warmup. DoRA leverages weight decomposition to enhance performance as a robust baseline. For FFT, we follow the settings proposed by corresponding datasets. For MoRA, we employed rotation operators as outlined in Eq. 9 to implement compression and decompression for $r = 8$, and for $r = 256$, we utilized shared rows and columns as specified in Eq. 6 and group every adjacent $\hat{r}$ rows or columns together. The details hyperparameters about fine-tuning can be found in Appendix A.

### 5.2.3 RESULTS AND ANALYSIS

We present the results of fine-tuning tasks in Table 2. We report the results of MMLU with zero-shot and 5-shot settings for instruction tuning, GSM8K and MATH for mathematical reasoning, and average performance on biomedical tasks and financial tasks for continual pretraining.

MoRA perform on par with LoRA in instruction tuning and mathematical reasoning. Benefit from high-rank updating to memorize new knowledge, MoRA outperforms LoRA on both biomedical and financial domains for continual pretraining.

We also find that LoRA variants exhibit similar performances on these fine-tuning tasks as compared to LoRA. Although AsyLoRA achieves the best performance in instruction tuning, it demonstrates poor performance in mathematical reasoning. For ReLoRA, merging low-rank matrices during training can harm performance, particularly at the the high rank like 256.

Consider the difference between three tasks, they show different requirements for fine-tuning capabilities. For instruction tuning, which does not learn new knowledge from fine-tuning, rank 8 is enough to achieve performance similar to FFT. For mathematical reasoning, rank 8 is unable to match FFT performance. However, increasing the rank from 8 to 256 can eliminate the performance gap. For continual pretraining, LoRA with rank 256 still underperforms FFT.

### 5.3 PRETRAINING

To understand the influence of high-rank updating, we train transformer from scratch on the C4 datasets Raffel et al. (2020). For the model architecture, we use LLaMA-based model with RM-

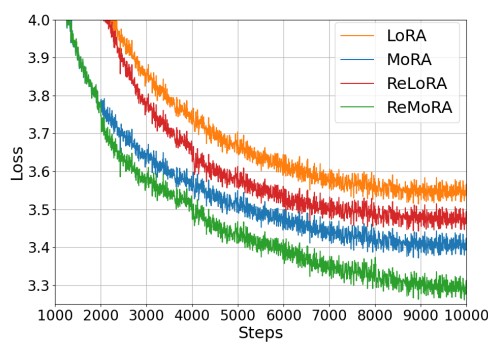

(a) Pretraining loss at 250M models.

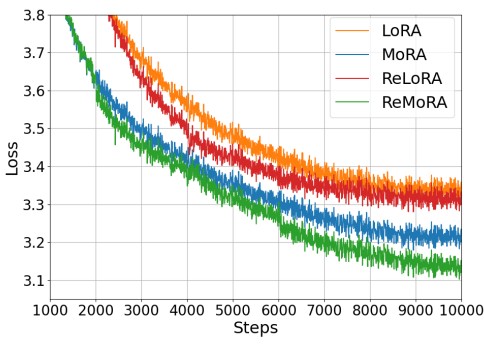

(b) Pretraining loss at 1.3B models.

Figure 4: Pretraining loss with LoRA and MoRA on 250M and 1B models from scratch. Both LoRA and MoRA use same amount of trainable parameters with $r = 128$. ReMoRA and ReLoRA refer to merge MoRA or LoRA back to the model during training to increase the rank of $\Delta W$.

SNorm Zhang & Sennrich (2019), SwiGLU Shazeer (2020) and RoPE Su et al. (2024), testing two sizes: 250M and 1.3B parameters. For the hyperparameters, we use 10k steps, 1024 batch size, 512 sequence length and applying rank $r = 128$ for LoRA and our methods follow Lialin et al.. We also leave modules layernorm or embeddings, which do not apply LoRA, unfrozen. To better show the difference between high-rank and low-rank updating, we reproduce LoRA and other methods without full-rank training warmup. For MoRA, we use Eq. 6 as compression and decompression functions by sharing columns and rows.

We also combine merge-and-reint in ReLoRA with our method called ReMoRA by merging $M$ back into the original parameters during training to increase the rank of $\Delta W$. However, if we directly merge $M$ with $g$ and $g'$ in Eq. 6, the final rank of $\Delta W$ is unchanged due to the same expand pattern. To solve this problem, we can change $g$ and $g'$ after merging to ensure the rank of $\Delta W$ increasing. More details about ReMoRA can be found in Appendix B. For the hyperparameters corresponding to ReLoRA and ReMoRA, we merge every 2k steps and use 50 steps restarts warmup with optimizer reseting and jagged scheduler.

We show pretraining loss in Figure 4 and corresponding perplexity on C4 validation dataset in Table 3. Our method show better performance on pretraining compared to LoRA and ReLoRA with same amount of trainable parameters. Benefiting from high-rank updating, ReMoRA also achieves more improvements on MoRA compared to ReLoRA, which demonstrates the effectiveness of merge-and-reint strategy in ReMoRA.

|  | 250M | 1.3B |
|---|---|---|
| LoRA | 33.40 | 28.56 |
| MoRA (Ours) | 28.54 | 25.25 |
| ReLoRA | 32.19 | 27.80 |
| ReMoRA (Ours) | 26.74 | 23.34 |

Table 3: Perplexity on C4 validation dataset.

# 6 ANALYSIS

## 6.1 INFLUENCE OF DECOMPRESSION AND COMPRESSION

To explore the impact of decompression and compression functions in MoRA, we report the performance on GSM8K using various methods: truncation, sharing, decoupling, and rotation in Table 4. Among these methods, truncation shows the worst performance due to the significant information loss during compression. Sharing can achieve better performance than truncation by leveraging the shared rows or columns to preserve the input information. But in the case of $r = 8$, sharing shows worse performance than decouple and rotation due to the large number of sharing rows or columns, as we discussed in Section 4. Rotation is more efficient than decouple, due to the rotation information can help the square matrix to distinguish the input information.

| | $f_{comp}, f_{decomp}$ | $r = 8$ | $r = 256$ |
|---|---|---|---|
| Truncation | Eq. 4 | 59.5 | 66.6 |
| Sharing | Eq. 6 | 62.5 | 67.9 |
| Decouple | Eq. 8 | 63.6 | 67.8 |
| Rotation | Eq. 9 | 64.2 | 67.9 |

Table 4: Influence of decompression and compression functions on $r = 8$ and $r = 256$ on GSM8K.

### 6.2 HIGH-RANK UPDATING

To demonstrate the impact of high-rank updating on the rank of $\Delta W$, we analyzed the spectrum of singular values for the learned $\Delta W$ on 250M pretraining 250M model. We present the average count of singular values exceeding 0.1 across all layers for $\Delta W_q$, $\Delta W_k$, $\Delta W_v$, $\Delta W_o$, $\Delta W_{up}$, $\Delta W_{down}$, and $\Delta W_{gate}$ in Figure 5 following Lialin et al. (2023). MoRA and ReMoRA exhibit a substantially higher number of significant singular values compared to LoRA and ReLoRA, highlighting the effectiveness of our methods in increasing the rank of $\Delta W$. We find the quantity of singular values shown in Figure 5 can be correlated with the perplexity metrics listed in Table 3. Moreover, MoRA, without merge-and-reint strategy in ReLoRA and ReMoRA, can achieve a lower perplexity than ReLoRA along with a higher significant singular values.

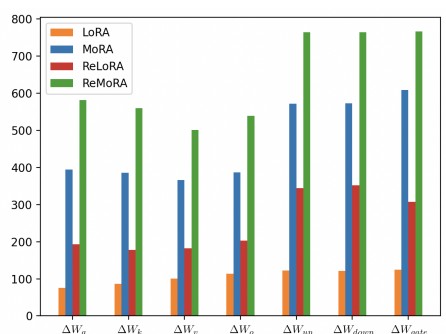

Figure 5: The number of singular values >0.1 in $\Delta W$ on the 250M pretraining model.

### 6.3 TRAINING SPEED AND MEMORY USAGE

Regarding training time and GPU memory usage, we benchmark LoRA and MoRA on the same hardware. For training settings, we run these methods on a single GPU with a sequence length of 1024, applying LoRA and MoRA to all linear layers of the 7B parameter model. The results are reported in Table 5. For $r = 256$, MoRA uses almost the same time and memory as LoRA, benefiting from the non-parameterized operators. Interestingly, we find that MoRA uses even less GPU memory than LoRA. However, for $r = 8$, MoRA employs Eq. 9 to compress and decompress input features, making it approximately 1.15 times slower than LoRA during fine-tuning.

| | Training Speed | Memory Usage |
|---|---|---|
| | $r = 8$ | |
| LoRA | 1.92 | 16.0GB |
| MoRA | 1.67 | 16.0GB |
| | $r = 256$ | |
| LoRA | 1.56 | 31.9GB |
| MoRA | 1.54 | 31.8GB |

Table 5: Comparison of training speed (steps/second) and memory usage with LoRA and MoRA with rank 8 and 256.

## 7 CONCLUSION

In this paper, we analyze the impact of low-rank updating through LoRA and observe it struggles with memory-intensive tasks, which also limits the performance of current LoRA variants. To overcome this limitation, we introduce MoRA, a method that utilizes non-parameterized operators for high-rank updating. Within the MoRA framework, we explore various methods to implement decompression and compression functions. Performance comparisons indicate that MoRA matches LoRA in instruction tuning and mathematical reasoning, and exhibits superior performance in continual pretraining and memory tasks. Additionally, we conduct pretraining experiments to further demonstrate the effectiveness of high-rank updating and show superior results compared to ReLoRA.

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

## A HYPERPARAMETERS

We report the hyperparameters in Table 6. The hyperparameters for Tülu v2 and MetaMath are following their papers Yu et al. (2023); Ivison et al. (2023). Additionally, we search for the optimal learning rate for LoRA across different tasks and report the best performance. We are able to reproduce the results in Yu et al. (2023); Ivison et al. (2023) with LoRA and even achieve better performance. For the hyperparameters of MoRA, we remove the $\alpha$ parameter from LoRA and use the same hyperparameters, except for the learning rate. The learning rate selection for MoRA with Eq. 6 may differ from that of LoRA. Due to the shared rows and columns in Eq. 6, MoRA exhibits a larger gradient norm, so we employ a smaller learning rate.

| Dataset | Method | $r$ | $\alpha$ | LR | LR Scheduler | Warmup | Epochs | Batch size | $f_{comp}$, $f_{decomp}$ |
|---|---|---|---|---|---|---|---|---|---|
| | FFT | - | - | 2e-5 | cosine | 500 | 2 | 128 | - |
| | LoRA-like | 8 | 16 | {1e-4,2e-4,3e-4} | cosine | 500 | 2 | 128 | - |
| Tülu v2 | MoRA | 8 | - | {1e-4,2e-4,3e-4} | cosine | 500 | 2 | 128 | Eq. 9 |
| | LoRA-like | 256 | 128 | {1e-4,2e-4,3e-4} | cosine | 500 | 2 | 128 | - |
| | MoRA | 256 | - | {3e-5,5e-5,7e-5} | cosine | 500 | 2 | 128 | Eq. 6 |
| | FFT | - | - | 2e-5 | cosine | 300 | 3 | 128 | - |
| | LoRA-like | 8 | 16 | {1e-4,2e-4,3e-4} | cosine | 300 | 3 | 128 | - |
| MetaMath | MoRA | 8 | - | {1e-4,2e-4,3e-4} | cosine | 300 | 3 | 128 | Eq. 9 |
| | LoRA-like | 256 | 128 | {1e-4,2e-4,3e-4} | cosine | 300 | 3 | 128 | - |
| | MoRA | 256 | - | {3e-5,5e-5,7e-5} | cosine | 300 | 3 | 128 | Eq. 6 |
| | FFT | - | - | 3e-5 | linear | 150 | 1 | 128 | - |
| | LoRA-like | 8 | 16 | {3e-4,4e-4,5e-4} | linear | 150 | 1 | 128 | - |
| BioMed./Fiance | MoRA | 8 | - | {3e-4,4e-4,5e-4} | linear | 150 | 1 | 128 | Eq. 9 |
| | LoRA-like | 256 | 128 | {3e-4,4e-4,5e-4} | linear | 150 | 1 | 128 | - |
| | MoRA | 256 | - | {5e-5,7e-5,1e-4} | linear | 150 | 1 | 128 | Eq. 6 |

Table 6: Hyperparameters for fine-tuning on three datasets.

## B IMPLEMENTATION OF REMORA

We introduce detial implementation of ReMoRA in pretraining. In this case, we simply define two kinds of $g$. The first kind is grouping every adjacent $\hat{r}$ rows or columns together following the defined in fine-tuning, the first groups can be represented as $\{1, 2, \ldots, \hat{r}\}$. The second kind is grouping every neighboring $k$ of the rows or columns together, the first groups can be represented as $\{1, 1+k, \ldots, 1+\hat{r}k\}$. We propose a example code about compression and decompression functions in Algorithm **1** and **2**. After merging, we can change the group type from 0 to 1 or 1 to 0.

---

**Algorithm 1** Compression

---

1: **function** COMPRESS($x$, $\hat{r}$, $type$)
2:     # $x \in \mathbb{R}^{bsz \times l \times k}$: Input tensor
3:     # $y \in \mathbb{R}^{bsz \times l \times \hat{r}}$: Output tensor
4:     # $type \in \{0, 1\}$: Group type 0 or 1
5:     padding $x$ to make $k$ divisible by $\hat{r}$
6:     **if** $type = 0$ **then**
7:         $y = x.view(bsz, l, k/\hat{r}, \hat{r}).sum(dim=2)$ # first type of group
8:     **else**
9:         $y = x.view(bsz, l, \hat{r}, k/\hat{r}).sum(dim=3)$ # second type of group
10:     **end if**
11:     **return** $y$
12: **end function**

---

## C DOWNSTREAM TASKS OF CONTINUAL PRETRAINING

For biomedcine, we use PubMedQA Jin et al. (2019), RCT Dernoncourt & Lee (2017), USMLE Jin et al. (2021), and selecting biomedicine subjects from MMLU to evaluate the performance. For finance, following BloombergGPT Wu et al. (2023),we use ConvFinQA Chen et al. (2022), NER Salinas Alvarado et al. (2015), Headline Sinha & Khandait (2021), FiQA SA Maia et al. (2018) and FPB Malo et al. (2014). We report the detail performance of these tasks following:

**Algorithm 2** Decompression

```
1: function DECOMPRESS(x, r̂, type)
2:     # x ∈ ℝ^{bsz×l×r̂}: Input tensor
3:     # y ∈ ℝ^{bsz×l×d}: Output tensor
4:     # type ∈ {0, 1}: Group type 0 or 1
5:     if type = 0 then
6:         y = repeat(x, d/r̂, dim=2) # first type of group
7:     else
8:         y = repeat-interleave(x, d/r̂, dim=2) # second type of group
9:     end if
10:    truncate y to ℝ^{bsz×l×d}
11:    return y
12: end function
```

| | $r$ | PubMedQA | USMLE | BioMMLU | RCT | Avg. |
|---|---|---|---|---|---|---|
| FFT | - | 74.1 | 41.2 | 47.5 | 62.7 | 56.4 |
| LoRA | 8 | 73.1 | 34.9 | 45.3 | 54.9 | 51.9 |
| MoRA | 8 | 73.3 | 34.7 | 45.3 | 59.9 | 53.3 |
| LoRA | 256 | 73.8 | 39.7 | 46.0 | 56.9 | 54.1 |
| MoRA | 256 | 74.4 | 40.4 | 46.1 | 60.6 | 55.4 |

Table 7: Performance on biomedical tasks.

| | $r$ | ConvFinQA | FiQA SA | Headline | NER | FPB | Avg. |
|---|---|---|---|---|---|---|---|
| FFT | - | 44.4 | 78.8 | 82.3 | 68.1 | 74.3 | 69.6 |
| LoRA | 8 | 44.5 | 76.2 | 72.4 | 61.6 | 65.1 | 64.0 |
| MoRA | 8 | 45.8 | 76.6 | 76.3 | 68.9 | 68.2 | 67.1 |
| LoRA | 256 | 41.4 | 78.3 | 83.0 | 66.8 | 66.7 | 67.3 |
| MoRA | 256 | 47.7 | 76.3 | 83.4 | 68.0 | 68.1 | 68.7 |

Table 8: Performance on finicial tasks.

