# OpenReview forum: "MoRA: High-Rank Updating for Parameter-Efficient Fine-Tuning"
_ICLR.cc/2025/Conference — ICLR 2025 Conference Withdrawn Submission_

### Official Review · Reviewer_jDnW · 2024-10-23

**Soundness:** 2
**Presentation:** 2
**Contribution:** 2
**Rating:** 5
**Confidence:** 4

**Summary:**

The authors suggest that low-rank updating mechanisms, as employed by LoRA, may limit the ability of large language models (LLMs) to effectively learn and memorize new knowledge, particularly in memory-intensive tasks. To address this limitation, they propose increasing the rank of the update matrix used in LoRA by employing a square matrix with a larger rank, while keeping the total number of trainable parameters constant. To achieve this, the authors introduce non-parametric operations that compress the input and decompress the output, ensuring dimensional compatibility for matrix multiplication. Additionally, they propose a rotation operation to preserve positional encodings during input compression, maintaining compatibility with matrix products.

**Strengths:**

The authors observe that LoRA’s low-rank updates (e.g., using a rank as low as 8) may not fully leverage the model's learning potential. To address this, they propose restructuring LoRA updates by compressing the input dimension, increasing the rank of the update matrix, and then decompressing the output to match the original dimensions. This approach maintains the same number of trainable parameters while increasing the rank, which is an interesting idea. However, while this method enhances learning capabilities in terms of memorization, the performance improvements on downstream tasks do not directly translate as significantly.

**Weaknesses:**

1. Unclear Compression and Decompression Operations: The compression and decompression operations described in equations (8) and following are not clearly explained. The rationale behind choosing these specific operations is unclear and requires further elaboration.

2. Rotation Operation: While the rotation operation is likely intended to preserve positional information after compressing the input, the explanation provided in the paper is not sufficiently clear. The authors need to clarify why rotation is added and how it enhances the matrix's ability to distinguish input information. The current description, especially the statement "rotation information can help the square matrix to distinguish the input information," is vague and needs more technical justification.

3. Performance Discrepancy in Table 1: In Table 1, MoRA with rank 8 achieves 100% accuracy, whereas LoRA with the same rank only achieves 52%. This discrepancy suggests that the LoRA model may have been undertrained or improperly tuned. The paper should address this discrepancy to ensure fair comparisons between methods.

4. Limited Improvement Across Tasks: MoRA shows clear improvements only in continual pre-training tasks, with little or no improvement in other downstream tasks. This suggests that MoRA is a task-specific method, which contradicts the goal of parameter-efficient fine-tuning methods that are expected to generalize across various tasks. This task-specific nature makes it challenging to justify its broad application without further experimentation, which defeats the purpose of an efficient fine-tuning method.

5. Uncertain Advantage of Rotation: The paper mentions that rotation helps the square matrix distinguish input information, but this benefit is not clearly demonstrated, particularly in higher-rank scenarios. At rank 256, the performance differences between the Sharing, Decouple, and Rotation methods are minimal. This raises the question of whether rotation is truly necessary at higher ranks, and the paper should clarify under what conditions (if any) rotation provides a real advantage.

6. For table 5, could you also provide the number of trainable parameters for both the methods under both ranks?

**Questions:**

Please see above.

---

### Official Review · Reviewer_YB3U · 2024-10-29

**Soundness:** 3
**Presentation:** 2
**Contribution:** 2
**Rating:** 3
**Confidence:** 3

**Summary:**

This paper proposes a novel PEFT method for LLM, which leverages squared matrix to achieve updates with higher ranks compared to baseline methods such as LoRA. It reduces the input dimension and increases the output dimension to match those in the square matrix with a novel strategy. Experimental results across five downstream tasks show that the proposed method outperforms LoRA on memory-intensive tasks and achieves comparable performance on other tasks.

**Strengths:**

1. This method, MoRA, is well-motivated by identifying weaknesses in LoRA and other low-rank adaptation methods. An interesting and intuitive experiment on memorizing UUID pairs further highlights the necessity of developing methods to address this limitation.

2. Experimental results on Continual Pretraining demonstrate MoRA's effectiveness in this scenario.

3. The provided code is also well-written.

**Weaknesses:**

1. The primary advantage of MoRA lies in tasks requiring significant enhancement of LLM knowledge and capabilities, such as memorizing UUID pairs and continual pretraining (CP). However, it may be more practical to directly use full finetuning in such cases, particularly for CP, which generally demands high computational resources to achieve optimal performance. This raises questions about the suitability of applying PEFT in such high-cost scenarios. As shown in Table 2, MoRA cannot outperform full FT in CP tasks. Additionally, on downstream tasks outside of CP, MoRA does not exhibit a clear advantage over baseline methods and generally results in decreased performance compared with standard LoRA, which limits its practicality.

2. In the final update matrix $\Delta W$ obtained with MoRA, many entries are zero. This could be a drawback, as some of these parameters might be essential for specific downstream tasks, yet MoRA is unable to update them. This limitation potentially restricts MoRA's effectiveness for LLM updates across a diverse range of downstream tasks.

3. The method section could benefit from improved organization, especially as there are no flowcharts or diagrams to visually illustrate the MoRA method. Currently, certain lines, such as the definitions of $f_{comp}$ and $f_{decomp}$ are somewhat confusing and could be clarified for better comprehension.

**Questions:**

Please refer to the points outlined in the weaknesses section.

---

### Official Review · Reviewer_RZh6 · 2024-11-02

**Soundness:** 3
**Presentation:** 2
**Contribution:** 2
**Rating:** 3
**Confidence:** 4

**Summary:**

This paper proposes a new parameter-efficient fine-tuning method, namely MoRA, designed to achieve high-rank updates to enhance the performance of large language models (LLMs) in memory-intensive tasks (tasks that require acquiring new knowledge for LLMs).

**Strengths:**

1. The paper provides a detailed analysis of the limitations of LoRA in tasks that require memorizing new knowledge.
2. The proposed high-rank updating method shows superior performance in memorization tasks compared to standard LoRA.

**Weaknesses:**

1. In sec 5.1 and sec 5.3, MoRA is only compared with standard LoRA, more baselines(e.g., [1][2]) are needed to substantiate your hypothesis that high-rank updating benefits memory-intensive tasks.
2. It appears that the experiments in Section 5.2 are aimed at demonstrating MoRA's superiority in memory-intensive tasks and its comparable performance on other tasks. However, results indicate that MoRA underperforms other baselines in mathematical reasoning, which I consider memory-intensive, as increasing the rank from 8 to 256 significantly improves the performance of both LoRA and MoRA.
3. The writing needs significant refinement:

 *  Line 160: "we employ LLaMA-2 7B as base model, utilizing 1,000 pairs per batch and conducting 100 epochs".
 * Line 165: "we observe low-rank updating are hard to memorizing new knowledge compared to FFT".
 * Line 191: "enhances the output dimension".

[1] AdaLoRA: Adaptive Budget Allocation for Parameter-Efficient Fine-Tuning

[2] DyLoRA: Parameter-Efficient Tuning of Pre-trained Models using Dynamic Search-Free Low-Rank Adaptation

**Questions:**

1. In sec 5.1, the search range of learning rate is from {5e-5,7e-5,1e-4}, does the results in Fig 3 are under the same learning rate setting?
2. Why are the baseline methods included in Section 5.2 excluded from Sections 5.1 and 5.3?

**Details Of Ethics Concerns:**

There are no concerns with this submission.

---

### Official Review · Reviewer_bjBX · 2024-11-11

**Soundness:** 4
**Presentation:** 3
**Contribution:** 3
**Rating:** 8
**Confidence:** 4

**Summary:**

This paper develops an alternative to LoRA that uses a square matrix to achieve a high rank update for an equivalent number of trainable parameters. They show that their approach is competitive with LoRA using Llama 2 7B across various training regimes (and datasets), spanning instruction finetuning, continual pretraining, and even pretraining from scratch. This method seems quite promising and is grounded in strong empirical results.

**Strengths:**

The authors motivate their approach by a clever memorization task. They show that vanilla LoRA with low rank (e.g. rank 8) does poorly on a contrived memorization task. They train on 10k unique IDs for 100 epochs where the LLM has to memorize the value associated with a key. This is a nice way to motivate the rest of the results.

The authors do rigorous benchmarking across training approaches, from instruction finetuning (IFT) and continual pretraining to pretraining from scratch. This is not so common among LoRA-style papers. The authors themselves bring up a valid critique that many LoRA-related papers focus on BERT style models and GLUE, which is not so relevant in the age of open source LLMs like Llama.

Very helpful to show the various effects of decompression approaches (Table 4)
Important to show the effect of training speed and memory usage - many LoRA studies leave this out.

This paper includes a final comparison of training latency and memory usage. Their method shows improvement over LoRA for the same number of trainable parameters, and is only 1.15x slower during training in certain settings.

**Weaknesses:**

1. The authors note that “increasing rank alleviates this problem” of memorizing UUID pairs. Although MoRA converges faster than LoRA at rank 256, they both end up converging. What are the benefits of using MoRA  rank 8 vs. LoRA rank 256? According to Table 2, they get roughly the same performance. If MoRA is a strict pareto improvement over LoRA for the same number of parameters, the paper should state so unambiguously (the paper states this in a few separate places for specific tasks). If there are any caveats these should be stated clearly as well.

2. There is one recent study that is quite similar to MoRA: LoRA-XS (first appeared May 2024) “LoRA-XS: Low-Rank Adaptation with Extremely Small Number of Parameters” (Bałazy et al.) https://arxiv.org/abs/2405.17604. This is very similar to the approach taken by MoRA, and it should be cited as contemporaneous work. Another related (but less similar) study is “VeRA: Vector-Based Random Matrix Adaptation” (Kopiczko et al. 2023) https://arxiv.org/pdf/2310.11454

3. For different datasets, it would be very helpful to compare the total number of samples and tokens. The authors make a distinction between IFT with few samples (e.g. LIMA) and more samples, but don’t give enough detail to compare continual pretraining. How large are the biomedicine and finance corpora? The LoRA Learns Less and Forgets Less paper (https://arxiv.org/abs/2405.09673) makes a distinction between IFT and continual pretraining based on the total number of tokens (they find LoRA with rank 256 does worse than full finetuning when trained on >5B tokens). Are these data regimes similar?


### Minor comments:
What is the hardware use for the speed/memory usage? This would be informative for Table 5. Otherwise it is somewhat meaningless
The authors don’t define continual pretraining; presumably this means that the corpus is used to train with next token prediction.
A handful of paragraphs suffer from poor/confusing grammar

Typos:

130: Compare → Compared

200: f_comp → f_decomp

256:use → uses

321: adapt to the → adapt to

342: 5.2.2 Baselines and Implements → Implementations

350: we search

355: The details parameters

362: grammar

369: grammar

399: leave modules layernorm → leave layernorm modules

**Questions:**

Why does MoRA rank 8 converge but LoRA rank 8 does not converge?

Important that the comparison with LoRA is with the same number of trainable parameters. How many total extra parameters does MoRA require, including the frozen compression and decompression stages?

---

### Note · Authors · 2024-11-18

**Comment:**

We would like to thank our sincere gratitude to the reviewers and Area Chairs for their invaluable feedback and insights. After careful consideration, we have decided to withdraw our submission. Thank you once again for your time and thoughtful evaluation.

**Withdrawal Confirmation:**

I have read and agree with the venue's withdrawal policy on behalf of myself and my co-authors.